# Triply Biobased Thermoplastic Composites of Polylactide/Succinylated Lignin/Epoxidized Soybean Oil

**DOI:** 10.3390/polym12030632

**Published:** 2020-03-10

**Authors:** Jianbing Guo, Jian Wang, Yong He, Hui Sun, Xiaolang Chen, Qiang Zheng, Haibo Xie

**Affiliations:** 1Department of Polymer Material and Engineering, College of Materials and Metallurgy, Guizhou University, Guiyang 550025, China; guojianbing_1015@126.com (J.G.); wangjian0319jian@163.com (J.W.); hy18798074906@163.com (Y.H.); 2National Engineering Research Center for Compounding and Modification of Polymer Materials, Guiyang, Guizhou 550014, China; chenxl612@sina.com; 3School of Materials Science and Mechanical Engineering, Beijing Technology and Business University, Beijing 100048, China; sunhui@th.btbu.edu.cn; 4Key Laboratory of Advanced Materials Technology Ministry of Education, School of Materials Science and Engineering, Southwest Jiaotong University, Chengdu 610031, China; 5Department of Polymer Science and Engineering, Zhejiang University, Hangzhou 310027, China

**Keywords:** lignin, polylactide composite, epoxidized soybean oil, dynamic vulcanization

## Abstract

Soybean oil is beneficial to improve the compatibility between polylactide (PLA) and succinylated lignin (SAL), which leads to the preparation of a host of biobased composites containing PLA, SAL, and epoxidized soybean oil (ESO). The introduction of SAL and ESO enables the relatively homogeneous morphology and slightly better miscibility obtained from triply PLA/SAL/ESO composites after dynamic vulcanization compared with unmodified PLA. The rigidity of the composites is found to decline gradually due to the addition of flexible molecular chains. According to the reaction between SAL and ESO, the *T*_g_ of PLA/SAL/ESO composites is susceptible to the movement of flexible molecular chains. The rheological behaviors of PLA/SAL/ESO under different conditions, i.e., temperature and frequency, exhibit a competition between viscidity and elasticity. The thermal stability of the composites displays a slight decrease due to the degradation of SAL and then the deterioration of ESO. The elongation at break and notched impact strength of the composites with augmentation of ESO increase by 12% and 0.5 kJ/m^2^, respectively. The triply biobased PLA/SAL/ESO composite is thus deemed as a bio-renewable and environmentally friendly product that may find vast applications.

## 1. Introduction

Lignocellulosic biomass is deemed as one of the most encouraging renewable and sustainable resources to lessen the increasing fuel demands [1,2,3,4], the growing concern for the effects of greenhouse gas emissions from fossil fuels, and the increasing accumulation of non-degradable waste from the high consumption of petroleum-based materials in many applications [5,6,7,8]. In particular, lignin is regarded as one of the main components of lignocellulosic biomass, which is composed of three biopolymeric compounds, namely cellulose, hemicellulose, and lignin, taking advantage of its unique property as the only aromatic polymer existing in the nature [9]. At present, lignin is deemed as a by-product of the pulp and paper industry, where it is typically recovered as waste material during pulping treatments. The limited exploitation of such a widespread renewable resource appeals to spending more time researching the most important efficient approaches to enhance lignin utilization as a precursor for the design and production of materials in view of the abundant aromatic structures. The application and development of lignin will contribute to the boom of bio-composites [10,11].

Lignin, a highly irregular, amorphous, and cross-linked biopolymer [12,13], is composed of three phenylpropanoid units, namely *p*-coumaryl alcohol (H units), coniferyl alcohol (G units), and sinapyl alcohol (S units), and various functional groups [14,15,16,17], which constitute a highly polar, complex, and heterogeneous biomacromolecule with a large quantity of hydroxyl (-OH) by the random permutation of units and groups that are linked to the major interunit linkages including β-O-4, α-O-4, β-5, β-β, 5-5, β-1, and 4-O-5 [18,19]. The diversity and complexity of lignin and the interconnected structure give rise to the low degree of compatibility and reactivity. Nevertheless, the chemical modification of lignin is a cost-effective way to improve the compatibility by increasing the intermolecular covalent bonds. Obviously, it is a convenient and inexpensive way to produce promising polymers by blending lignin with other polymers. However, developing the blend of petroleum-based polymer with lignin may accelerate the depletion of fossil-based reserves and deterioration of the environment. In light of this consideration, it is promising to search for potential bio-renewable/biodegradable polymers to replace traditional synthetic polymers. In this context, poly(lactide) (PLA) is considered to be the first choice for blending with lignin. It is anticipated that a new kind of biodegradable and biocompatible bio-composites will be developed.

PLA has been chosen because this polymer is derived from corn starch and sugarcane biomass and is known as a promising bio-polymer with outstanding biodegradability and biocompatibility [20,21]. Currently, extensive attention is given to PLA since it is considered as a potential replacement for petroleum-based sources with renewability and good mechanical properties. However, its high cost and poor impact strength hinder the wide-ranging commercial applications of the polymer [22]. PLA endowed with modified lignin is expected to yield inexhaustible composites having relatively good compatibility between the matrix and filler. Unexpectedly, the toughness of PLA/lignin composites is expected to be improved by polymer blending, owing to the hindrance of the intrinsic brittleness of PLA and lignin itself.

Plant oils are an inexpensive, green, and sustainable feedstock to toughen PLA/lignin composites in comparison with other renewable polymers [23], such as biomass-sourced polyesters, natural rubber and its derivatives, and biobased polyamide [24,25,26,27,28,29]. Castor oil and soybean oil have been considered as a toughening agent for PLA [30,31]. In order to obtain good immiscibility between plant oils and PLA, the phase morphology and interfacial adhesion are constituted by dynamic vulcanization [32,33,34,35].

Traditionally, dynamic vulcanization is a process where the rubber is vulcanized in the presence of the molten thermoplastic under shear forces and the rubber particles are dispersed in the thermoplastic matrix, even at a high content, in order to improve the ultimate mechanical properties, reduce the permanent tension, improve the fatigue resistance, and so on [36]. In addition, it is found that many polymer blends/composites with high mechanical properties are fabricated by using dynamic vulcanization, which is a very powerful way to control the phase morphology, enhance the interfacial adhesion, and improve the tensile and impact strengths of polymers [32,34,35]. The objective of this work is to fabricate new PLA composites with modified lignin and epoxidized soybean oil (ESO) of different concentrations by dynamic vulcanization [37,38] and to elucidate the relationship between the structure and property of such composites. By adjusting the volume of ESO during the process of blending, fully sustainable and renewable PLA/succinylated lignin (SAL)/ESO composites are obtained to probe the interfacial interaction, miscibility, and compatibilities of the materials, for the sake of accomplishing widespread application in the future, such as bio-composites and functional materials.

## 2. Materials and Methods

### 2.1. Materials

Poly(lactic acid) (PLA 4032D) with a weight-average molecular weight (Mw) and a polydispersity index of 17.62 × 10^4^ g/mol and 2.1 was procured from Natureworks (Blair, NE, USA). Alkaline lignin was supplied by Shandong Quanlin Paper Co., Ltd. (Gaotang, China), and was subject to H_2_SO_4_ pretreatment. Soybean oil epoxide (ESO) with a density of 0.997 g/mL was purchased from MACKLIN. Gamma-valerolactone (GVL), 4-dimethylaminopyridine, succinic anhydride (SA), and isopropyl ether were purchased from Aladdin Co., Ltd. (Shanghai, China)

### 2.2. Synthesis of Succinylated Lignin Adducts

Lignin (3.0 g) was dissolved in GVL (2.5 g) to form a homogeneous solution under magnetic stirring. Then, the solution was stirred for 5 h at 140 °C to activate the lignin. Next, 3.0 g succinic anhydride (SA) and 0.15 g 4-dimethylaminopyridine were added into the mixture and stirred for another 1 h at 100 °C. The crude reaction solution was added dropwise to the isopropyl ether. The yielded SAL precipitate was centrifuged, washed thoroughly with deionized water, and freeze-dried before further utilization [39]. The synthesis route of succinylated lignin is presented in Scheme 1.

### 2.3. Dynamic Vulcanization of ESO and SAL with PLA

The dynamic vulcanization of ESO and SAL with PLA was performed in a HAAKE PolyLab QC torque rheometer (Haake co., Karlsruhe, Germany). At first, SAL and PLA were placed in an oven at 80 °C for 8 h. Then, PLA, SAL, and ESO were uniformly mixed according to the mass ratio (listed in Table 1) in the torque rheometer at 190 °C with a roller rotation rate of 60 rpm for 8 min. After, the predetermined mixture cooled to room temperature and was cut into granules, for the preparation of the composites.

### 2.4. Preparation of Composites

The prepared granules were placed in an oven at 100 °C for 8h, and the mixed materials were extruded on a micro twin-screw extruder (SJZS-10A, Ruiming Experiment Apparatus Manufacturing Co., Ltd., Wuhan, China) with a main screw speed of 160 r/min and a feeding speed of 8 r/min, at heating temperatures of 180, 190, and 200 °C. A micro-injection machine (SZS-20, Ruiming Experiment Apparatus Manufacturing Co., Ltd., Wuhan, China) was used to injection mold the compound into a spline at 190 °C according to ASTM D-790. The injection molded samples were then placed in a thermostatic biochemical chamber (temperature 23 °C, humidity 80%) for 24 h to eliminate internal stress. In addition, the blend was injection molded into a disc having a diameter of 2 mm and a thickness of 2 mm by extrusion molding for dynamic rheological testing.

### 2.5. Characterization and Measurements

#### 2.5.1. Fourier Transform Infrared Spectroscopy

FTIR spectra were recorded on a NEXUS-570 spectrophotometer (Thermo Fisher Scientific Co., Ltd., Waltham, MA, USA) over a wavenumber range of 4000 to 500 cm^−1^ with a resolution of 4 cm^−1^, and a scanning number of 32 times. Each sample was ground, mixed with KBr at a mass ratio of about 1/100, and pressed into pellets before FTIR analysis.

#### 2.5.2. Nuclear Magnetic Resonance Spectroscopy

NMR spectra were recorded on a Bruker Ascend 400 MHz spectrometer. The lignin samples were phosphitylated with 2-chloro-4, 4, 5, 5-tetramethyl-1, 3, 2-dioxaphospholane, using cholesterol as the internal standard, according to the standard protocol before ^31^P NMR analysis. The experimental parameters, i.e., scans, delay, and spectral width, were 128, 15s, and 80 ppm (180−100 ppm), respectively. Quantitative ^1^H NMR analysis was recorded on a Bruker Ascend 400 MHz spectrometer (Bruker Corporation, Karlsruhe, Germany) at room temperature using DMSO-d6 as the solvent, with 32 scans and a delay of 10 s. The ^13^C-NMR spectra were recorded on a Bruker Ascend 400 MHz spectrometer (Bruker Corporation, Karlsruhe, Germany) using an inverse-gated decoupling sequence with a relaxation time of 4 s and a number of scans of at least 10,000. The samples were dissolved in DMSO-d6 and measured at 60 °C.

#### 2.5.3. Dynamic Mechanical Analysis 

The rectangular samples of the composites were studied by a TA Instruments DMA (Q800, TA Co., Ltd., New Castle, PA, USA) under tensile mode. Frequency sweep was performed over a range from 5 to 45 Hz, at 30 °C, with a vertical displacement amplitude of 15 μm, to obtain dynamic mechanical properties.

#### 2.5.4. Differential Scanning Calorimetry

DSC measurements of about 8 mg sample were carried out on a TA Instruments Q10 (TA Instruments, TA Co., Ltd., New Castle, PA, USA). The samples were heated from room temperature to 200 °C and kept there for 5 min to eliminate thermal history. The samples were then quenched to 30 °C and finally reheated to 200 °C for the study of the crystallization and melting behavior of the composites.

#### 2.5.5. Thermogravimetric Analysis 

TGA **measurement** of samples of 10–50 mg **was** recorded on a TA Instruments Q50 (TA Co., Ltd., New Castle, PA, USA) under nitrogen gas flow of 10 mL/min, with a heating rate of 10 °C /min and a preset temperature of 800 °C.

#### 2.5.6. Rheological Analysis

The rheological properties were measured using a TA rheometer (AR 2000, TA Co., Ltd., New Castle, PA, USA) with a nitrogen purge. A 25 mm parallel-plate with a 1000 μm gap was selected for the test. A strain amplitude of 1% was found to be suitable to ensure the linear viscoelastic regime and was thus used for both the frequency sweep and temperature ramp sweep modes.

#### 2.5.7. Mechanical Properties

Tensile tests were measured with an electronic universal mechanical testing machine (WDW-10C, Shanghai Hualong Test Instrument Co., Ltd., Shanghai, China) according to ASTM D-638. The experimental parameters were 10 mm/min crosshead speed at room temperature and 25 mm gauge length between the two pneumatic clamps. The average of five measurements was reported in the analysis of the mechanical properties.

#### 2.5.8. Scanning Electron Microscopy 

The morphologies of the PLA/SAL samples were observed by Quanta FEG230 scanning electron microscope (FEI Co., Ltd., Hillsboro, OR, USA) equipment with an acceleration voltage of 25 kV. SEM images were recorded after the surfaces of the samples were coated with gold.

## 3. Results and Discussion

### 3.1. The Characterization of SAL Adducts

The esterification of lignin was recorded by means of FTIR spectroscopy. As shown in Figure 1, SAL exhibited a signal characteristic of the occurrence of the esterification between lignin and succinic anhydride. The wide absorption band at 3429 cm^−1^ originated from O-H stretching vibrations in aromatic and aliphatic O-H groups, whereas the bands at 2942 and 2845 cm^−1^ were attributed to the C-H asymmetric and symmetric vibrations in the methyl and methylene groups. The peaks at 1595 and 1520 cm^−1^ were ascribed to the C-C of the aromatic skeletal vibrations. Bands at 1460 and 1420 cm^−1^ were linked to the C-H deformation in the –CH_2_- and –CH_3_ groups and C-H aromatic ring vibrations, respectively. Some characteristic bands found at 1325, 1220, and 1110 cm^−1^ corresponded to syringyl and condensed guaiacyl absorptions, guaiacyl ring breathing, C-C, C-O stretching, and aromatic C-H in-plane deformation. Non-conjugated carboxylic acids were observed at the 1730 cm^−1^ band. In the case of lignin and SAL in Figure 1, it was observed obviously that the signal around 3401 cm^−1^ corresponding to aromatic and aliphatic OH stretching vibration was enhanced compared with lignin. The intensified carboxylic groups at 1730 cm^−1^ in SAL were associated with more carboxylic groups. The assignments and analyses of the peaks proved the successful esterification.

The successful modification of lignin with SA was also confirmed by NMR analysis, as shown in Figure 2. Obviously, the broad proton signal at δ 6.0–7.8 were attributed to aromatic protons, while the resonance peaks at δ 4.8–3.0 were assigned to the methoxy group in the aliphatic/aromatic region. Proton signals from hydroxyl groups were easily affected by hydrogen bonding, leading to the uncertainty in the changing trend of hydroxyl groups. By comparison, after functionalization with SA, it was deemed that the COOH groups of SAL were increased and the corresponding OH groups were transformed to form COOH groups. The signals of -CH_2_ or -CH_3_ in SAL were found to be enhanced owing to the occurrence of a reaction. The comparative ^13^C NMR spectra of lignin and SAL in Figure 3 indicated that the increased signals at 160–190 ppm of C=O groups in SAL samples were attributed to the new formation of ester and COOH groups. In addition, new d signals appearing at 30–20 ppm in SAL samples were assigned to the form of side chains.

The reaction of lignin with SA was anticipated to increase the carboxylic acid groups and decrease the hydroxyl groups in lignin. In this regard, quantitative ^31^P NMR spectroscopy was shown to be a powerful method to evaluate the functional groups distribution in lignin and SAL. As shown in Figure 4 and Table 2, after modification of lignin with SA, the total hydroxyl group content was increased from 4.67 mmol/g to 6.31 mmol/g. Theoretically, the total hydroxyl groups in lignin would not change after the modification as the consumption of one mole of aliphatic or aromatic hydroxyl group would generate one mole of the carboxylic acid hydroxyl group. This increases possibly accounted for the separation of lignin with low hydroxyl groups during the modification and subsequent purification process. The values of aliphatic hydroxyl groups and aromatic hydroxyl groups in SAL were lower than those in the raw lignin, implying low chemical selectivity to the chemical modification of aliphatic and aromatic hydroxyl groups in lignin. Notably, the value of the carboxyl acid hydroxyl groups in the SAL was up to 4.5 mmol/g, much higher than the 0.54 mmol/g for the raw lignin, further indicating the successful chemical reaction between the hydroxyl groups in lignin with SA.

### 3.2. Processing and Rheological Behaviors

Several comprehensive reviews already dealt with the basic influencing factors of the torque value [40], which is analogous to the apparent viscosity (η_a_) of materials and contingent on the molecular weight, as well as the chain structure of the polymer. The torque curves of all samples during dynamic vulcanization are displayed in Figure 5. The torque of pristine PLA showed a monotonous decrease with increasing the mixing time, which was attributed to the partial thermal degradation of PLA chains during processing [41]. Compared with neat PLA, PLA/ESO composites exhibited a lower equilibrium torque value. Furthermore, the torque values of PLA/SAL/ESO composites with 5 wt% SAL showed a decline with increasing the content of ESO. This indicated that the degradation of PLA mainly led to the change of the torque value due to the addition of SAL and ESO. It was also observed that the torque values of PLA/SAL/ESO composites gradually decreased with increasing the ESO content, which indicated the formation of short chains or branching chains in the reaction system. However, the torque curves of PLA/SAL/ESO composites displayed a new peak when the dosage of ESO exceeded 2%. This observation may be explained by the reaction between carboxylic groups of SAL and epoxy groups of ESO, which led to either crosslinking or branching structures, as shown in Scheme 2.

Figure 6 shows the storage modulus (*G*′), loss modulus (*G*″), and complex viscosity (*η**) of pure PLA and PLA/SAL/ESO composites at 170 °C. It is noted that the *η** value decreased gradually with increasing frequency, indicating that both PLA and PLA/SAL/ESO composites belonged to the typical pseudoplastic fluid. Another observation was that the values of *G*′ and *G*″ were enhanced progressively with increasing frequency and showed a relatively smooth trend eventually, which brought it into correspondence with the linear viscoelasticity theory of polymers. At low frequency, the variation of shear force was slow, and the movement of molecular chains emulated the variation of stress, which led to the viscosity of materials owing to the high *G*″ value. However, at high frequency, the increasing rate of *G*′ changed fast, and *G*′ attained a higher value than that of *G*″. As a consequence, the movement of the molecular chain was inconsistent with the change in stress, causing the material to show elasticity rather than viscosity. The result extended the knowledge that the *G*′, *G*″, and *η** values of PLA/SAL/ESO composites were lower than those of pure PLA, which resulted from the lower viscosity of composites and the relatively weak interface effect and coefficient of friction [42].

Figure 7 shows the temperature ramp sweep results of pure PLA and PLA/SAL/ESO composites. It is seen from this Figure 7 that the values of *G*′ of all PLA/SAL/ESO composites decreased drastically with increasing temperature from 180 to 210 °C, which was attributed to the transform from a rubbery plateau to a flow behavior. In the temperature range of 210 to 240 °C, the presence of a slow decrease was observed due to the complete melting of PLA. However, the values of *G*′ of pure PLA and PLA/ESO composites were found to decline at a certain ratio with increasing frequency. It was revealed that *G*′ for all the PLA/SAL/ESO composites was lower than that of pure PLA and PLA/ESO. This resulted from the formation of flexible molecular chains and the better interfacial interaction and miscibility. The above observations clearly showed that the *G*′ of PLA/SAL/ESO composites was more sensitive to the low temperature range. In addition, the *G*″ and *η** values of all samples were found to decrease dramatically with increasing temperature from 180 to 240 °C. This indicated that PLA composites gradually turned into a viscous state. Nevertheless, the values of *G*″ were still higher than those of *G*′ at a specific temperature, indicating the viscous flow of PLA rather than elastic deformation.

### 3.3. Dynamic Mechanical Analysis

The storage modulus (*E*′), loss modulus (*E*″), and tanδ of neat PLA and its composites are shown in Figure 8. It is seen from Figure 8a that pure PLA showed higher *E*′ values than PLA/ESO and PLA/SAL/ESO composites, indicating the higher rigidity of pure PLA. It was also found that the value of *E*′ of PLA/SAL/ESO composites gradually decreased with the addition of ESO, suggesting that the rigidity of the composites gradually declined because of the plasticization of ESO and the flexibility of the molecular chains.

Particular emphasis is placed on understanding the *E*″ concept, which stands for the degradation or the loss of energy as heat according to the cycle of sinusoidal distortion, as shown in Figure 8b. It appeared that the of pure PLA showed a lower *E*″ value than the other samples at a certain temperature range, which demonstrated that the motion of the polymer happened to be transformation. This testified further that the viscosity of the PLA/SAL/ESO composites was higher than that of PLA. As a consequence of the addition of SAL and ESO, the interfacial interaction between PLA and SAL with ESO was improved due to the plasticizing effect of ESO.

The tanδ curves displayed in Figure 8c also reveal the comparison of the glass transition temperature (*T*_g_). Both the PLA/SAL/ESO and PLA/ESO composites exhibited lower *T*_g_ than pure PLA, which was ascribed to the addition of SAL and ESO. A decline in *T*_g_ was also found for PLA/SAL/ESO with increasing the loading of ESO due to the poor interfacial interaction between PLA and SAL caused by the addition of ESO and the poor dispersion of homogeneous particles. At the same time, the compatibility between the polymer matrix and fillers in the PLA/SAL/ESO composites became slightly worse, as a consequence of the limited interfacial interaction.

### 3.4. Melting and Crystallization Behaviors

Complete details of the materials are provided in Figure 9 to probe the interaction between the SAL, ESO, and PLA phase on the glass transition temperature (*T*_g_) and melting temperature (*T*_m_). The *T*_g_ value of PLA/ESO composites became lower than that of pure PLA, due to the plasticizing effect of ESO on the composites. When both SAL and ESO were added to PLA, it was discovered that the onset *T*_g_ of PLA/SAL/ESO composites gradually shifted to a lower temperature with increasing ESO concentration, compared to that of pure PLA. On the basis of the results, it was concluded that molecular level reciprocity between the blending components triggered the variation of *T*_g_ in the blend composition [43]. Owing to the reaction between SAL and ESO, which improved the compatibility of the three components, SAL, ESO, and PLA, the increase of molecular pliability caused the phenomenon. In addition, ESO could act as a plasticizer to improve the mobility of molecular chains, leading to a decrease of the *T*_g_ value.

The *T*_m_ was a remarkable indicator that enabled the analysis of phase miscibility of PLA, SAL, and ESO [44]. As shown in Figure 9 and Table 3, an identical consequence was obtained in this work where the *T*_m_ value of PLA/SAL/ESO composites was impaired slightly compared with pure PLA. The reduction in *T*_m_ was primarily ascribed to the deterioration of the crystalline phase of PLA affected by the molecular miscibility with SAL and ESO. With the addition of SAL and ESO, the triply composites gradually became more amorphous due to the complexity of SAL and ESO.

### 3.5. Thermal Stability

The TGA profiles of PLA, PLA/ESO, and PLA/SAL/ESO composites are displayed in Figure 10 and Table 4. It is clearly observed that the thermal stability of the composites gradually deteriorated with increasing the ESO content from 0 to 6 wt%, according to the data of *T*_5%_ and *T*_max_. PLA/ESO composites had a similar thermal behavior. Inevitably, it was found that the PLA/SAL/ESO composites showed poorer thermal stability than PLA and PLA/ESO, possibly resulting from the degradation of molecular chains, when the compatibility region between PLA, ESO, and SAL was poor [45]. With the addition of SAL, the residue of PLA/SAL/ESO composites was higher than those of PLA and PLA/ESO composites. However, different PLA/SAL/ESO composites displayed a similar trend, indicating that the amount of ESO played a weak role in the thermal stability of the materials. Concerning the char residues, the presence of SAL was responsible for the increase of the carbon-based residues of materials, and SAL was a charring material only under pyrolytic conditions [46]. When the SAL existed in the composites, the char formation together with the strong interaction resulted in the enhancement of residues.

### 3.6. Mechanical Properties

The mechanical properties were also analyzed, and the stress-strain curves are displayed in Figure 11. It was found that the corresponding Young’s modulus of PLA/SAL/ESO composites was higher than that of pure PLA and PLA/ESO composites, which was ascribed to the addition of SAL, endowing rigidity to PLA/ESO composites owing to its aromatic nature. It is worthwhile to mention that PLA/ESO composites had a higher Young’s modulus than pure PLA due to the interaction between PLA and ESO. As expected, it was found in the stress-strain curves that the Young’s modulus of PLA/SAL/ESO composites gradually decreased with increasing ESO concentration, When the addition of SAL was constant, ESO acted as a toughening agent to improve the mechanical properties of PLA/SAL/ESO composites to overcome the intrinsic brittleness of PLA and SAL with increasing the content of ESO [47]. It is pointed out that the yield point of PLA/SAL/ESO composites was found to decline compared with pure PLA and PLA/ESO composites. In other words, the mechanical properties of PLA/SAL/ESO composites were reduced with increasing the ESO loading due to the existence of SAL particles, which prevented the formation of a long range continuous phase of PLA [48]. Table 5 shows that the tensile strength, flexural strength, and flexural modulus exhibited a trend of declining with increasing the content of ESO. However, the elongation at break and notched impact strength increased gradually with the augmentation of ESO. It is seen from Figure 11 that the elongation at break of PLA/lignin/ESO composites was lower than those of PLA/SAL/ESO composites, which was attributed to the poor compatibility between PLA and lignin in the presence of ESO. Overall, SAL played a great role on the SAL particles dispersing in PLA with the addition of ESO.

SEM is essential in probing particle size and interfacial interaction. The micrographs of pure PLA, PLA/ESO, and PLA/SAL/ESO composites are shown in Figure 12. Some particles and cracks were uniformly dispersed on the surface of PLA [49], leading to a weaker molecular interaction. Figure 12a,g reveals that the addition of ESO improved the dispersion of PLA and interfacial interaction with PLA, indicating that ESO had a positive impact on the miscibility of PLA. However, higher ESO loading gave rise to wire drawing of the surface in PLA/ESO composites. It is seen that PLA/lignin/ESO composites showed a clear interface with some holes and fiber pulling, which illustrated the weaker interfacial adhesion between lignin and PLA owing to the addition of unmodified lignin in Figure 12h. When it came to the excessive addition of SAL to PLA/ESO composites, it was found that PLA/SAL/ESO composites exhibited better interfacial morphology with increasing the ESO content at constant SAL loading. Compared with pure PLA, the composites had a smaller particle size and smoother surface, as shown in Figure 12b–d. The reason was attributed to the reactive compatibilization between the carboxyl (-COOH) of SAL and epoxy groups of ESO, which improved the properties of PLA/SAL/ESO composites by dynamic vulcanization. At a 4% ESO volume, PLA/SAL/ESO composites showed a relatively homogeneous morphology with a much smoother surface, demonstrating a superior interfacial interaction. When the ESO loading was higher than 4%, unreacted ESO was regarded as a reinforcing agent to improve the interface interaction and miscibility of PLA/SAL/ESO composites owing to the formation of a smaller cavity from the SAL particles [50]. Although some holes were observed, they played a supporting role for the mechanical properties of composites to relieve the deformation of the materials.

## 4. Conclusions

Fully sustainable PLA/SAL/ESO composites were fabricated by dynamic vulcanization of PLA/SAL/ESO precursors. The morphology and properties of the PLA/SAL/ESO composites were dependent on the interfacial interaction and miscibility of SAL determined by ESO. The phase morphology changed from phase-separation for pure PLA behaving relatively homogeneously for PLA/SAL/ESO3 (PLA: SAL: ESO3=0.91: 0.05: 0.04, wt%). The compatibility was enhanced and the dispersed particle size on SAL and ESO declined with increasing the content of ESO. At a high concentration of ESO, the formation of smaller holes resulted in a rougher interface of the phase, but the cavity played a supporting role for the materials by relieving the effect of stress. Obviously, there was no doubt that the rigidity of PLA/SAL/ESO composite gradually declined due to the addition of flexible molecular chains. The PLA/SAL/ESO composites was susceptible to the movement of flexible molecular chains and the weaker interfacial interaction with increasing the content of ESO, which was ascribed to the high concentration of ESO, which was considered as a reinforcing agent.

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
