# Peer review of "Triply Biobased Thermoplastic Composites of Polylactide/Succinylated Lignin/Epoxidized Soybean Oil"

_polymers, 2020, doi:10.3390/polym12030632_

Round 1

Reviewer 1 Report

The study « Triply bio-based thermoset composites of polylactide/succinylated lignin/epoxidized soybean” oil by J. Guo et al. aims to combine Lignin and PLA to produce triply PLA/SAL/ESO a sustainable and renewable composite. For this purpose Lignin is modified into succinylated lignin (SAL) and mixed with epoxidized soybean oil (ESO) and PLA. The subject is of interest and the compounds used are promising in the field. The experimental work is of good quality. Thus, the paper could be accepted after minor revisions listed below:

“The carboxylic acid functional groups of high content in SAL will provide active sites for reaction with epoxides in ESO to form cross-linked structures”. Does the authors have evidence of this with TFIR or NMR measurements to show creation of chemical bonds?

Why torque measurements were used instead of dynamic viscosity measurements?

Line 258: “It is concluded that the compatibility and interfacial adhesion between PLA and SAL are enhanced with increasing the content of ESO”  and Line 268: “As a consequence of addition of modified SAL and ESO, the interfacial interaction between PLA and SAL with ESO is improved due to the plasticizing effect of ESO.”

In my opinion conclusions on compatibility and adhesion should be more argued  to improve the quality of the discussions of the results. Conclusions between E’’and tan delta seems contradictory? Are the conclusions similar between DMA, stress-strain measurements and SEM ?

Author Response

Thanks for the valuable comments from the referees and editorial office to improve the manuscript. Herein, we submit our response to the remarks accordingly. Furthermore, we have improved the language through the whole manuscript and the main changes were highlighted in red color.

1.“The carboxylic acid functional groups of high content in SAL will provide active sites for reaction with epoxides in ESO to form cross-linked structures”. Do the authors have evidence of this with TFIR or NMR measurements to show creation of chemical bonds?

 Response: After the additional investigations and discussions, excepting the NMR results in Fig 4, it is lack of more powerful evidence to prove those “provided active sites”. The description has been revised in manuscript.

2. Why torque measurements were used instead of dynamic viscosity measurements?

Response: The torque measurements are not used instead of dynamic viscosity measurements. However, in this work, the processability of all samples is characterized by the torque measurement, which can find the peak during the reactive process to prove the existence of crosslinking. In addition, the rheological behaviors including viscosity are characterized by using a TA rheometer, and the corresponding results are shown Figures 6 and 7.

3. Line 258: “It is concluded that the compatibility and interfacial adhesion between PLA and SAL are enhanced with increasing the content of ESO” and Line 268: “As a consequence of addition of modified SAL and ESO, the interfacial interaction between PLA and SAL with ESO is improved due to the plasticizing effect of ESO.”

Response: We have revised the two sentences of the same meaning to avoid the redundant description in our revision.

4. In my opinion conclusions on compatibility and adhesion should be more argued to improve the quality of the discussions of the results. Conclusions between E’’and tan delta seems contradictory? Are the conclusions similar between DMA, stress-strain measurements and SEM ?

Response: Thank you very much for your good comments.The lower tan delta values mean that the materials are more close to the ideal elastic materials. There are no such contradictory between E” and tan delta because the values of tan delta are obtained form E’ and E”. The similar conclusions are also drawn in analysis of static mechanical properties and SEM.

Reviewer 2 Report

The manuscript describes a biobased composites composed of PLA, SAL and epoxidized soybean oil (ESO). The subject is actual.  Lignin is the most abundant natural polyphenol aromatic polymer. Biobased composites should present better biodegradability and an enhanced eco-friendliness. However, the manuscript need improvements before being accepted for publication.  

One of the needed improvement is the envisaged application. Authors should discuss in  Section 3. Result and discussion the advantages of  decreased rigidity of the  PLA/SAL/ESO composite for the envisaged application.  General and vague sentence like that from the L77, Introduction section “for the sake of accomplishing widespread application in the future” shall be not considered a substitute for presentation of concrete potential application of the resulted biobased composite.

L34-L41. First paragraph of the Introduction Section shall be re-written. This paragraph aims to present lignin as a by-product of the biomass biorefinery. However, this is not clear enough.

L63-L65. The phrase is not clear and shall be refined. “Designed” meaning is  “done intentionally; intended; planned”. It is a logical break between this “intentional done” and “Unexpected” word from the beginning of the phrase.

According to Template of Polymers Section 2 shall be entitled  Materials and Methods and not Experimental.

Subsection  2.1. Materials. More data related to the used biopolymer shall be presented- e.g. Melt Point or Glass Transition Temp for PLA and at least sulphur content for lignin. Precise description of the materials is necessary for other researcher to reproduce the results. There are no data regarding the epoxidized soybean oil (ESO). Beside the producer, of ESO, technical characteristics critical for the application (production of modified lignin – PLA – ESO composites) shall be included (e.g.  FlashPoint or Ester content)

L80. Polylactide (PLA 203D) probably refers to Ingeo™ 2003D. Probably the company is not  NatureWorks and not America Nature. In order to avoid confusion it is necessary to present not only company name, but also the city/town where the company is located and the state/country – i.e. NatureWorks, Minnetonka, MN, USA. In the whole section Material and methods, name of the material suppliers and equipment producer shall be mentioned, together with the city/town where the company is located and the state/country.

L92 Scheme 1. Please complete with “ in the presence of 4-dimethylaminopyridine”.

Subsection 2.3. Preparation of composites. Name of the producer of micro twin-screw extruder / micro-injection machine (SJZS-10A type) is not included. Code of the Standard  for test methods for flexural properties of  composites (ASTM D-790) shall not be in brackets. More details regarding the determination of flexural properties and torque calculation during composite preparation shall be briefly presented.  More detail about dynamic vulcanization and curing process are needed.

Subsection 3.2. Processing and rheological behaviors. The used method for the data presented here shall be included into Subsection 2.3. Preparation of composites.

L387-L509. Please avoid double numbering – adjust your  Reference editor with MDPI-_7_Reference Style.

L511-L536. The caption of Figures shall not be presented on the end of the manuscript.

Author Response

Thanks for the valuable comments from the referees and editorial office to improve the manuscript. Herein, we submit our response to the remarks accordingly. Furthermore, we have improved the language through the whole manuscript and the main changes were highlighted in red color.

1. The manuscript describes a biobased composites composed of PLA, SAL and epoxidized soybean oil (ESO). The subject is actual. Lignin is the most abundant natural polyphenol aromatic polymer. Biobased composites should present better biodegradability and an enhanced eco-friendliness. However, the manuscript need improvements before being accepted for publication.

Response: We have revised and discussed our manuscript carefully in our revision according to your comments.

2. One of the needed improvement is the envisaged application. Authors should discuss in Section 3. Result and discussion the advantages of decreased rigidity of the PLA/SAL/ESO composite for the envisaged application. General and vague sentence like that from the L77, Introduction section “for the sake of accomplishing widespread application in the future” shall be not considered a substitute for presentation of concrete potential application of the resulted biobased composite.

Response: We have revised all sections carefully in our revision according to your comments.

3. L34-L41. First paragraph of the Introduction Section shall be re-written. This paragraph aims to present lignin as a by-product of the biomass biorefinery. However, this is not clear enough.

Response: We have modified the First paragraph of the Introduction Section in the manuscript.

4. L63-L65. The phrase is not clear and shall be refined. “Designed” meaning is “done intentionally; intended; planned”. It is a logical break between this “intentional done” and “Unexpected” word from the beginning of the phrase.

 Response: We have modified the part in the manuscript.

5. According to Template of Polymers Section 2 shall be entitled Materials and Methods and not Experimental.

Response: We have modified the part in the manuscript.

6. Subsection 2.1. Materials. More data related to the used biopolymer shall be presented- e.g. Melt Point or Glass Transition Temp for PLA and at least sulphur content for lignin. Precise description of the materials is necessary for other researcher to reproduce the results. There are no data regarding the epoxidized soybean oil (ESO). Beside the producer, of ESO, technical characteristics critical for the application (production of modified lignin–PLA–ESO composites) shall be included (e.g. FlashPoint or Ester content)

Response: We have modified the part in the manuscript.

7. L80. Polylactide (PLA 203D) probably refers to Ingeo™ 2003D. Probably the company is not NatureWorks and not America Nature. In order to avoid confusion it is necessary to present not only company name, but also the city/town where the company is located and the state/country – i.e. NatureWorks, Minnetonka, MN, USA. In the whole section Material and methods, name of the material suppliers and equipment producer shall be mentioned, together with the city/town where the company is located and the state/country.

Response: We have modified the part in the manuscript.

8. L92 Scheme 1. Please complete with “in the presence of 4-dimethylaminopyridine”. Response: We have modified the part in the manuscript.

9. Subsection 2.3. Preparation of composites. Name of the producer of micro twin-screw extruder / micro-injection machine (SJZS-10A type) is not included. Code of the Standard for test methods for flexural properties of composites (ASTM D-790) shall not be in brackets. More details regarding the determination of flexural properties and torque calculation during composite preparation shall be briefly presented. More detail about dynamic vulcanization and curing process are needed.

Response: We have revised them in our revision according to your comments. In addition, more detail information of dynamic vulcanization and curing process have been provided in this revision.

10.Subsection 3.2. Processing and rheological behaviors. The used method for the data presented here shall be included into Subsection 2.3. Preparation of composites.

Response: We have revised them in our revision according to your comments.

11. L387-L509. Please avoid double numbering – adjust your Reference editor with MDPI-_7_Reference Style.

 Response: We have modified the double numbering in reference part in the manuscript.

12. L511-L536. The caption of Figures shall not be presented on the end of the manuscript.

Response: Revised as mentioned.

Reviewer 3 Report

Article Title: Triply bio-based thermoset composites of  polylactide/succinylated lignin/epoxidized soybean oil

In this work, PLA composites are prepared by dynamic vulcanization of PLA with modified lignin and epoxidized soybean oil (ESO) of different concentrations.

This work can be considered for publication after major revisions as follows:

  1. “Elongation at break and notched impact strength of the composites with augmentation of ESO show an increasing trend”. Here author need to provide quantitative statement in the abstract.
  2. What is dynamic vulcanization? That need to be explained in the manuscript (see the following ref.: Rubber Chemistry and Technology 90.1 (2017): 1-36)
  3. Authors have mentioned that this composite has widespread application in the future. Authors need to provide such suitable potential applications with justification in order to improve scientific quality of this paper.
  4. There is one mistake in the manuscript. In Figure 6, complex viscosity (η*) is plotted against frequency. Complex viscosity is a complex number from its storage and loss factor. The data authors plotted in Figure is not complex viscosity (η*), it is actually absolute value of complex viscosity (recommend to see this ref.: European Polymer Journal88, pp.221-230)
  5. Quality of the discussion have to be improved.
  6. There are several grammatical mistake. These have to be corrected and language of the manuscript should be edited. 

Author Response

Thanks for the valuable comments from the referees and editorial office to improve the manuscript. Herein, we submit our response to the remarks accordingly. Furthermore, we have improved the language through the whole manuscript and the main changes were highlighted in red color

  1. “Elongation at break and notched impact strength of the composites with augmentation of ESO show an increasing trend”. Here author need to provide quantitative statement in the abstract.

Response: We have added the part of abstract in our manuscript.

  1. What is dynamic vulcanization? That need to be explained in the manuscript (see the following ref.: Rubber Chemistry and Technology 90.1 (2017): 1-36)

Response: We have explained the dynamic vulcanization in our revision according to your comments.

  1. Authors have mentioned that this composite has widespread application in the future. Authors need to provide such suitable potential applications with justification in order to improve scientific quality of this paper.

Response: We have revised them in our revision according to your comments.

  1. There is one mistake in the manuscript. In Figure 6, complex viscosity (η*) is plotted against frequency. Complex viscosity is a complex number from its storage and loss factor. The data authors plotted in Figure is not complex viscosity (η*), it is actually absolute value of complex viscosity (recommend to see this ref.: European Polymer Journal, 88, pp.221-230)

Response: We have revised them in our revision according to your comments.

  1. Quality of the discussion have to be improved.

Response: We have checked and rewritten the discussion in this revision.

  1. There are several grammatical mistake. These have to be corrected and language of the manuscript should be edited.

Response: We have polished our manuscript carefully with the help of a professional professor. The wrongs including English language have been corrected in this revision.

Round 2

Reviewer 3 Report

Still one fundamental point is missing.

What is dynamic vulcanization? This has to define in the introduction section. 

Whether the actual concept of dynamic vulcanization is applicable here? This reviewer had suggested to see the following ref.: (Rubber Chemistry and Technology 90.1 (2017): 1-36).

Author Response

Thanks for the valuable comments to improve the manuscript. Comments: 1. What is dynamic vulcanization? This has to define in the introduction section. Whether the actual concept of dynamic vulcanization is applicable here? This reviewer had suggested to see the following ref.: (Rubber Chemistry and Technology 90.1 (2017): 1-36). Response: Revised as mentioned. Related words have been added into the main text (Line 79-87) as “Traditionally, dynamic vulcanization is a process where the rubber is vulcanized in the presence of the molten thermoplastic under shear forces and the rubber particles are dispersed in the thermoplastic matrix, even at high content, in order to improve ultimate mechanical properties, reduced permanent tension set, improved fatigue resistance and so on [36]. In addition, it is found that many polymer blends/composites with high mechanical properties are fabricated by using dynamic vulcanization, which is a very powerful way to control the phase morphology, enhance the interfacial adhesion, and improve the tensile and impact strengths of polymers [37-39]. The objective of this work is to fabricate new PLA composites with modified lignin and epoxidized soybean oil (ESO) of different concentrations by dynamic vulcanization [40, 41]”.